# All-small-molecule organic solar cells with over 14% efficiency by optimizing hierarchical morphologies

Ruimin Zhou[1,2,3,4,7], Zhaoyan Jiang[1,2,7], Chen Yang[1,2], Jianwei Yu[5], Jirui Feng[6], Muhammad Abdullah Adil[1,2], Dan Deng[1], Wenjun Zou[1], Jianqi Zhang [1], Kun Lu [1]*, Wei Ma[6]*, Feng Gao [5]* & Zhixiang Wei[1,2]*

The high efficiency all-small-molecule organic solar cells (OSCs) normally require optimized morphology in their bulk heterojunction active layers. Herein, a small-molecule donor is designed and synthesized, and single-crystal structural analyses reveal its explicit molecular planarity and compact intermolecular packing. A promising narrow bandgap small-molecule with absorption edge of more than 930 nm along with our home-designed small molecule is selected as electron acceptors. To the best of our knowledge, the binary all-small-molecule OSCs achieve the highest efficiency of 14.34% by optimizing their hierarchical morphologies, in which the donor or acceptor rich domains with size up to ca. 70 nm, and the donor crystals of tens of nanometers, together with the donor-acceptor blending, are proved coexisting in the hierarchical large domain. All-small-molecule photovoltaic system shows its promising for high performance OSCs, and our study is likely to lead to insights in relations between bulk heterojunction structure and photovoltaic performance.

[1] CAS key laboratory of nanosystem and hierarchical fabrication, CAS Center for Excellence in Nanoscience, National Center for Nanoscience and Technology, 100190 Beijing, China. [2] University of Chinese Academy of Sciences, 100049 Beijing, China. [3] Sino-Danish Center for Education and Research, Sino-Danish College, University of Chinese Academy of Sciences, 100190 Beijing, China. [4] Nano-Science Center and Department of Chemistry, University of Copenhagen, DK-2100 Copenhagen, Denmark. [5] Department of Physics Chemistry and Biology (IFM), Linköping University, SE-58183 Linköping, Sweden. [6] State Key Laboratory for Mechanical Behavior of Materials, Xi'an Jiaotong University, 710049 Xi'an, China. [7] These authors contributed equally: Ruimin Zhou, Zhaoyan Jiang. *email: lvk@nanoctr.cn; msewma@xjtu.edu.cn; feng.gao@liu.se; weizx@nanoctr.cn

Organic bulk heterojunction (BHJ) solar cells have attracted wide attention due to their advantages of lightweight, low cost, flexibility and compatibility with large-area printing fabrication[1–7]. Currently, considerable progress in the design and synthesis of efficient non-fullerene acceptors, as well as the device structure optimization has led to a rapid increase in the power conversion efficiency (PCE) of the organic solar cells (OSCs). Therefore, the PCE of the polymer solar cells (PSCs) with the non-fullerene acceptor has boosted to over 16%[8]. However, the batch-to-batch reproducibility of polymers still potentially limits their application on the industrial scale. In comparison with polymer donors, small-molecule donors possess the advantages of well-defined molecular weight, easy purification and small batch-to-batch variations[9–13]. Furthermore, enhanced crystallization, as well as their tendency to obtain high phase purity and tune crystal orientations enables fabrication of OSCs with high charge mobility and low energy losses ($E_{loss} = E_g - qV_{OC}$)[12,14,15]. Despite all their advantages, the all-small-molecule organic solar cells (SM-OSCs) still have not shown the same level of device performance as the PSCs. The biggest challenge is to control the interpenetrating networks in SM-OSCs, since the inefficient charge transport pathways would lead to excessive exciton recombination, decreased charge carrier mobility and unbalanced charge-transport ability[16–18]. Hierarchical morphology provides a potential strategy for a balanced charge separation and charge transport simultaneously in this regard[14,18–20]. Therefore, the design and synthesis of small-molecule donors that are matched well with the acceptor materials are vastly important for optimal morphology and efficient devices.

As a notable aromatic analog of benzodithiophene (BDT), dithieno[2,3-d:2′,3′-d′]benzo [1,2-b:4,5-b′]dithiophene (DTBDT) holds a larger coplanar core and an extended conjugation length, which can effectively improve the properties of photovoltaic materials compared to BDT-based molecules, such as the charge carrier transport, reduction in the conformational disorder of the backbone, and increase the molecular planarity to facilitate electron delocalization in the solid-state. Accordingly, stronger crystallinity and more ordered morphology in thin films can be realized[21–29]. Sun et al. reported a DTBDT-based polymer PDBT-T1 that exhibited a high FF of 75% due to the formation of optimized fibril network morphology[21], which is beneficial for efficient photo-generated exciton dissociation and charge collection. BDT units have been successfully incorporated into the small-molecule donors for both fullerene and non-fullerene solar cells[17,18,30–35], among which the highest PCE of 11.50% and 13.63% has been attained for non-fullerene based binary and ternary blends, respectively[30,36]. However, the DTBDT unit has seldom been introduced into the small-molecule donors in the literature[27–29], since blending them with non-fullerene acceptors could only yield a low PCE of 4.38%, as reported by Kwon et al.[28]. The exciton decay loss, that is driven by an inappropriate phase separation of the small molecules has been proved to be a crucial factor limiting photocurrent generation in the SM-OSCs. These results thus suggest that the ability to modulate the phase separation is the key factor to improve the photovoltaic performances of SM-OSCs.

In this work, we report a DTBDT-based small-molecule donor named ZR1 with an A-π-D-π-A architecture. The electron-rich DTBDT is utilized as the donor (D) unit to effectively extend the conjugated plane and improve the planarity of the molecule. Instead of the conventional trithiophene, bithiophene was chosen as a π bridge to deepen the HOMO level and to increase the rigidity unit ratio. Owing to the complementary and broad absorption, the optimized ZR1:Y6 based devices reached the highest PCE of 14.34% (certified PCE of 14.1%) and exhibited a low $E_{loss}$ of 0.52 eV. The single-crystal structure of ZR1 showed high planarity and a compact molecular packing with a short π−π

stacking, which was well matched with the molecular stacking in the ZR1:Y6 blend. Hence, the hierarchical morphology plays a key role in improving charge extraction efficiency and reducing charge recombination in the corresponding devices. Likewise, a non-fullerene acceptor IDIC-4Cl with good crystallinity was synthesized and blended with ZR1. The results showed an interesting hierarchical phase separation, demonstrating the largest and smallest domain size of 147.4 nm and 6.2 nm, respectively. The ZR1:IDIC-4Cl blend films exhibited a highest PCE of 9.64%, which is limited by the relatively higher $E_{loss}$ and wider bandgap of IDIC-4Cl as compared to Y6.

## Results

**Materials design and device properties.** The molecular structures of ZR1, IDIC-4Cl, and Y6 are shown in Fig. 1a, while the detailed synthesis, purification, and characterization procedures are provided in Supplementary Methods. The small-molecule donor ZR1 was synthesized through the Knoevenagel reaction with high yields of over 70% (Supplementary Fig. 1). Due to long side chains on its 2D-conjugated thiophene units and rhodanine end acceptor, ZR1 exhibited excellent solubility in chloroform. Hexyl substituted rhodanine was selected as the ending acceptor (A) group, as our previous result showed that such configuration leads to the highest performance[37]. A recently reported narrow bandgap small-molecule Y6 with absorption edge more than 930 nm, and a home-designed acceptor IDIC-4Cl were selected as the electron acceptors[38]. The non-fullerene acceptor IDIC-4Cl was also prepared through Knoevenagel condensation with a high yield. IDIC-4Cl and Y6 also exhibited good solubility in chloroform, which is beneficial for the fabrication of photovoltaic devices.

The normalized absorption spectra of ZR1, IDIC-4Cl and Y6 in chloroform solutions and as-cast thin films are shown in Fig. 1b. The absorption edge of the ZR1 donor is located at ca. 675 nm in the film, corresponding to a medium optical bandgap of 1.84 eV. Likewise, the absorption edge of Y6 and IDIC-4Cl acceptor films are located at ca. 930 nm and ca. 810 nm, corresponding to a relatively narrow optical bandgap of 1.33 eV and 1.53 eV, respectively. (Supplementary Table 1). Therefore, the ZR1 donor can effectively provide complementary absorption for both IDIC-4Cl and Y6 non-fullerene acceptors, which is beneficial to achieve high short circuit current density ($J_{SC}$). Importantly, a large bathochromic shift of 90 nm from solution to film is observed for Y6, indicating a good aggregation of the molecular backbone and π−π interactions in the solid-state.

Cyclic voltammetry (CV) measurements were employed to evaluate the electrochemical properties of the donor molecule and the acceptors(Supplementary Fig. 2)[39]. The onset oxidation and reduction potentials were used to calculate the HOMO and LUMO energy levels, and the absolute energy level of FeCp$^{+2/0}$ was set as −4.8 eV as a reference. The HOMO and LUMO energy levels of ZR1, IDIC-4Cl and Y6 were calculated to be at –5.32 eV and –3.53 eV, –5.72 eV and –4.10 eV, and –5.91 eV and –4.10 eV, respectively (Fig. 1c). Therefore, the ZR1 donor exhibited appropriate molecular energy level alignment with acceptors IDIC-4Cl and Y6, necessary for the efficient thermodynamic driving force for charge separation.

Conventional devices with a device architecture of ITO/poly(3,4-ethylenedioxythiophene): poly(styrenesulfonate)(PEDOT:PSS)/active layer/Al were fabricated to evaluate the photovoltaic performance of the ZR1 donor with IDIC-4Cl and Y6 as the acceptors. The current density-voltage (J–V) characteristics of the devices were tested under AM 1.5 G illumination with an intensity of 100 mW cm$^{-2}$. The ZR1: Y6 devices were certified at an accredited laboratory, producing a certifying PCE of 14.1% (Supplementary Fig. 3). Notably, all devices were fabricated without any additives

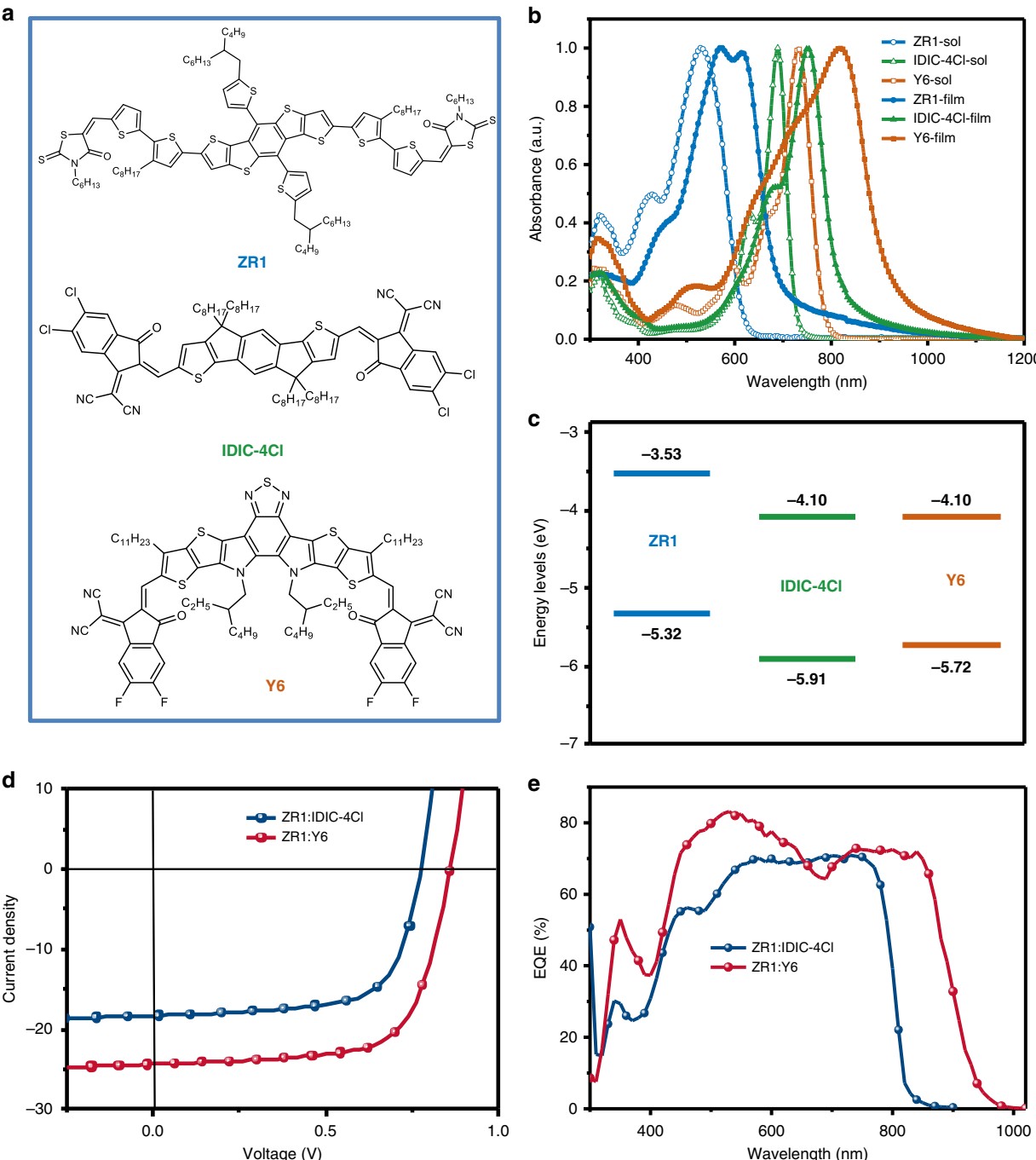

**Fig. 1** Molecular structures, properties and photovoltaic performance. **a** ZR1, Y6, IDIC-4Cl molecular structures. **b** Normalized UV–vis absorption spectra of ZR1, Y6, IDIC-4Cl in solution and thin films. **c** Energy diagrams of ZR1, Y6, IDIC-4Cl. **d** Optimized J–V curves for conventional devices. **e** EQE corresponding to devices in **d**.

and electron-transporting layer, making them an important prospect for future industrial manufacturing. The donor/acceptor ratios of the blending films were carefully optimized to improve the device performance. Thermal annealing (TA) treatment was a key factor for improving the morphology of the active layers and hence further increased the PCE. The ZR1:IDIC-4Cl blends exhibited the best photovoltaic performance with a D:A ratio of 1:0.7, followed by thermal annealing (TA) at 120 °C for 10 min. Without TA, the ZR1:IDIC-4Cl devices showed poor photovoltaic performance with low $J_{SC}$ and fill factor (FF). The highest PCE of ZR1:IDIC-4Cl blend film reached 9.64%, demonstrating an open-circuit voltage ($V_{OC}$) of 0.776 V, a $J_{SC}$ of 18.27 mA cm$^{-2}$ and a FF of 67.96% (Fig. 1d, and Table 1, while the device results under different D:A ratio are summarized in Supplementary Table. 2). Similarly, the optimized devices with ZR1 as donor and Y6 as acceptor attained a highest PCE of 14.34% with a $V_{OC}$ of 0.861 V, a $J_{SC}$ of 24.34 mA cm$^{-2}$ and a FF of 68.44%, while maintaining a D:A ratio of 1:0.5 and post-spin coating TA treatment at 120 °C for 10 min (Fig.1d). Altering the TA temperature to 110 °C and even 140 °C led to a reduction in device performance as a $V_{OC}$ of 0.870 V and 0.837 V, a $J_{SC}$ of 24.02 mA cm$^{-2}$ and 24.49 mA cm$^{-2}$, FF of 66.20% and 67.90% and a PCE of 13.62% and

**Table 1 Detailed photovoltaic parameters of the OPV cells.**

| Donor/acceptor | $V_{OC}$ [V] | $J_{SC}$ [mA cm$^{-2}$] | FF [%] | PCE [%] Best | Average[a] |
|---|---|---|---|---|---|
| ZR1:Y6 | 0.861 | 24.34 | 68.44 | 14.34 | 14.27 |
| ZR1:IDIC-4Cl | 0.776 | 18.27 | 67.96 | 9.64 | 9.58 |

[a]The average PCE values were obtained from top 10 devices

13.91%, respectively were observed. Without TA treatment, the ZR1: Y6 blends showed poor photovoltaic performance as a $V_{OC}$ of 0.876 V, a $J_{SC}$ of 14.23 mA cm$^{-2}$ and an FF of 40.5% was observed, ultimately leading to a PCE of 5.05% (Supplementary Table. 3,4). Likewise, the solvent vapor annealing (SVA) did not show a positive impact as compared to the solely thermal annealed devices, as a decrease in the $J_{SC}$ and FF was observed when the corresponding devices were solvent vapor annealed with THF (Supplementary Table 5). Two donor small molecules with monothiophene and trithiophene as π bridges, namely ZR1-T and ZR1-3T, respectively, were also synthesized for the sake of comparison. The molecule structure, UV-absorption spectrum, molecule energy levels and device performance are exhibited in Supplementary Figs. 4, 5, and Supplementary Table 6, respectively. The devices based on ZR1-T:Y6 blends exhibited extremely poor performance as the HOMO of ZR1-T and Y6 lie at almost the same energy levels, and hence cannot provide enough driving force for exciton dissociation and charge transport. For ZR1-3T blend, a relatively higher HOMO level led to a low $V_{OC}$ of 0.754 V as compared to 0.861 V for the ZR1 system. Likewise, the $J_{SC}$ and FF of ZR-3T: Y6 blends were also lower than ZR1-3T: Y6 blends, resulting in much lower efficiency than ZR1.

The EQE curves of these two systems were measured to explicate the discrepancy in the $J_{SC}$ values of the devices. The EQE spectra in Fig. 1e showed that both these systems exhibited broad absorption, but due to a wide absorption range from Y6, the ZR1: Y6 curve extended up to 950 nm. These results are identical with the UV-vis absorption spectra for the molecules. Meanwhile, the $J_{SC}$ values calculated from the EQE measurements were in accordance with the J–V results, as shown in Table 1. The considerable discrepancy in FF values of the two systems could be resulted by the different crystallinities of the three molecules and the distinguishing differences in morphologies of the blend films, which will be discussed in the next section.

**Molecular stacking of pristine and blend films**. Grazing incidence wide-angle X-ray scattering (GIWAXS) measurements were performed to study the molecular stacking and crystallinity of the pure and blend films. The 2D scattering patterns and intensity profiles in the out-of-plane (OOP) and in-plane (IP) direction are shown in Fig. 2a–g, and Supplementary Fig. 6, respectively. The small-molecule donor ZR1 shows an edge-on orientation demonstrated by the π–π peak located in the IP direction. Interestingly, after thermal annealing, the ZR1 exhibited strong molecular stacking with diffraction peaks at $q_z = 1.1$ to 1.8 Å$^{-1}$ that can be associated with the polycrystalline ordering of ZR1 at 1.71 Å$^{-1}$ with a π–π stacking distance of 3.67 Å. Single-crystal X-ray diffraction analysis was performed to further analyze the molecular structure and packing in their condensed state (Fig. 2h, i, and Supplementary Fig. 7). The single crystals of ZR1 were prepared by the way of slow diffusion of methanol (poor solvent) to their chloroform (good solvent) solution. The dihedral angles between the DTBDT core and π bridge came out to be 3.13° which indicated a high degree of planarity and rigidity in ZR1. Furthermore, ZR1 showed highly ordered and compact

molecular packing with a short π–π stacking distance with 3.58 Å, which would definitely improve hole transport. Similar to the most non-fullerene acceptors, pure Y6, IDIC-4Cl film exhibits face-on orientation indicated by the π–π peak located at $q_z = 1.75, 1.84$ Å$^{-1}$ ($d = 3.59$ Å, 3.41 Å), respectively.

It can be seen from the as-cast blend film that the π–π stacking peak area is located at 1.72 Å$^{-1}$, which is dominated by the peak of Y6, and the d-spacing is slightly larger than that of the neat thin film (Fig. 2c). The strong (010) peak in the out-of-plane direction indicates a preferable face-on orientation of molecules relative to the substrate. By increasing the TA temperature, the intensity of (100) and (010) peaks, as well as the ratio of (010) in the out-of-plane direction is also increased. Furthermore, for the TA temperatures of 120 °C and 140 °C, an additional peak at 0.59 Å$^{-1}$ in the IP and OOP direction is observed, indicating the presence of ZR1 crystals. A clear feature comprising of face-on and edge-on crystallites was observed in the thermally annealed films at 120 °C and 140 °C, suggesting the co-existence of vertical and parallel charge transportation channels in the sandwich device structure. In comparison, the pristine blend film of ZR1: IDIC-4Cl also exhibited face-on orientation with relatively weak peak intensities (Supplementary Fig. 3). Upon thermal annealing, significantly stronger peaks with narrower widths, accompanied by the presence of new peaks, such as the secondary and third lamellar peaks from donor component, were clearly discernible. Furthermore, the stronger and sharper lamellar (100) and (200) diffraction peaks of IDIC-4Cl appeared, indicating significantly enhanced crystallinity of the donor and acceptor as a consequence of thermal annealing.

The Scherrer equation was employed to calculate the crystal coherence lengths (CCL) of the crystallites[40]. The CCL (100) in the OOP direction of pure ZR1 film came out to be 7.31 nm and 10.89 nm, before and after TA, respectively, indicating improved crystallinity post thermal processing. Likewise, the CCL (100) of the ZR1:Y6 film in the OOP direction turned out to be 4.07 nm, 6.21 nm, 6.50 nm, 6.90 nm for the as-cast and films annealed at 110 °C, 120 °C, and 140 °C, respectively.

**Morphology analysis**. The morphological evaluation of the ZR1: Y6 and ZR1:IDIC-4Cl blends were investigated by using the transmission electron microscopy (TEM) and Resonant soft X-ray scattering (RSoXS) measurement (Fig. 3). Astonishingly, the primary domain sizes of ZR1:Y6 blends and ZR1:IDIC-4Cl were relatively large, which is normally believed to be a drawback for exciton diffusion and dissociation. After TA treatment, phase separation got even more enhanced as a consequence of even larger domain sizes. Moreover, increasing the TA temperature beyond 120 °C led to the formation of numerous aggregates with a large aspect ratio, which can be assigned to the ZR1 donor which is consistent with the GIWAXS measurements. Since the best performance transpires when the devices are thermally annealed at 120 °C, the presence of a small number of ZR1 crystals are indicative of facilitating the charge transport, while excessive crystals would lead to a reduction in the charge generation efficiency as a consequence of ineffective D:A interaction at the interface. ZR1:IDIC-4Cl blends, on the other hand, show fibrous features with more evenly distributed domains. Such fibrous morphology may be beneficial for the bi-continuous interpenetrating networks, essential for efficient charge transport and hence enabling relatively high $J_{SC}$ and FF.

RSoXS was employed for investigating the phase separation and phase purity of the blending films. The photon energy of 284.2 eV was selected to achieve the optimized material contrast. The large domain sizes for ZR1:Y6 based blends, for as-cast and films annealed at 110 °C, 120 °C, and 140 °C came out to be 73.4, 72.3, 73.9, and 83.4 nm, respectively (Supplementary Table 7),

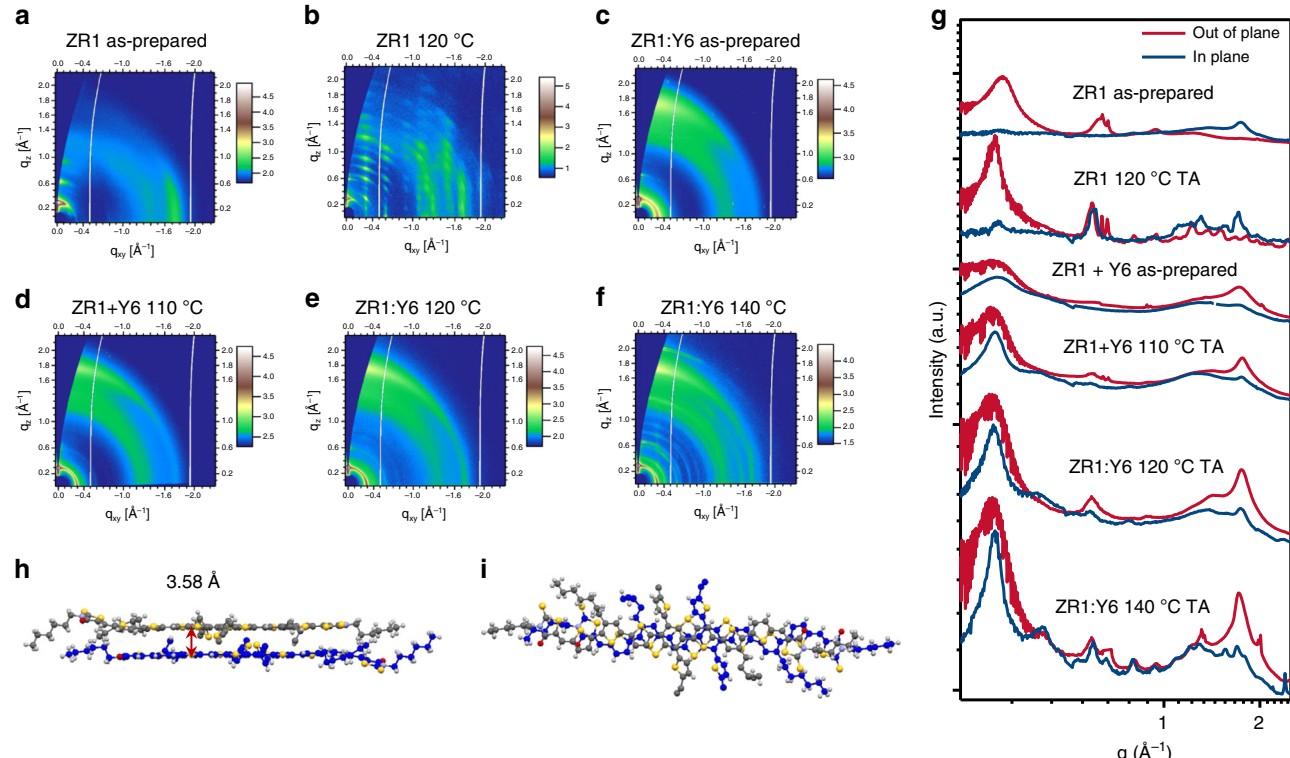

**Fig. 2** Microstructures of pristine and blend films. **a–f** 2D GIWAXS patterns of pristine and blend films. **g** Corresponding out-of-plane curves and in-plane curves. **h** View of perpendicular to π-stacking of dimers of ZR1 molecules in its single crystal and (**i**) along the π-stacking direction.

and exhibited a relatively increasing tendency along with increasing thermal temperature, which matches well with TEM results above. Moreover, the small phase area appeared to be coexisting at the same time for ZR1:Y6 based blend films, under as-cast, 110 °C, 120 °C, and 140 °C thermal annealing conditions, at a domain size ca. 10 nm. The relative domain purities of ZR1:Y6 blends for as-cast and films annealed at 110 °C, 120 °C, and 140 °C were calculated to be 0.87, 0.90, 0.95, and 1, respectively, and also exhibited an increase along with increasing thermal temperature. For ZR1:IDIC-4Cl films annealed at 120 °C, the domain size of the coexisting large and small phase area turned out to be 147.4 nm and 6.2 nm, respectively (Fig. 3h). These results suggest the formation of a hierarchical morphology within the blend i.e., a smaller donor phase, having a size very close to the exciton diffusion length of ca. 10 nm which accounts for the efficient charge separation, along with a larger donor phase that is responsible for the efficient charge transport within the system. In contrast to the previous reports about the SM-OSCs' hierarchical morphology[18], the donor crystals, about 100 nm long and 30 nm wide, were also observed. The presence of these optimized hierarchical morphologies indicated that the nano-structural characteristics with multiple length scales, as well as coexisting crystals are among key factors for high performance.

To understand the role of this hierarchical morphology, the charge transport properties were investigated for as prepared pristine and blend films, as well as for the blend films at the optimized device conditions. The space charge limited current (SCLC) method was employed to measure the mobility of the mentioned systems. A device architecture of (ITO/PEDOT:PSS/ active layer/MoOx/Ag) was maintained for examining the hole only devices, whereas for electron-only devices, an architecture of (ITO/ ZnO/Active layer/Al) was employed. The hole ($\mu_h$) mobilities of the ZR1: IDIC-4Cl and ZR1: Y6 films were calculated to be $3.30 \times 10^{-4}$ $cm^2 V^{-1} s^{-1}$ and $1.32 \times 10^{-4} cm^2 V^{-1} s^{-1}$, respectively, whereas

the electron ($\mu_e$) mobilities came out to be $4.55 \times 10^{-4} cm^2 V^{-1} s^{-1}$ and $3.92 \times 10^{-4} cm^2 V^{-1} s^{-1}$, respectively. (Supplementary Table 8). Thus, these results reflect that the hierarchical morphologies of both systems are efficient for charge transport.

**Energy loss**. According to the Shockley–Queisser (SQ) limit, the energy loss ($q\Delta V_{OC}$) in solar cells can be sorted into three parts [41]:

$$
\begin{aligned}
q\Delta V_{OC} = E_g - qV_{OC} &= \left( E_g - qV_{OC}^{SQ} \right) \\
&\quad + \left( qV_{OC}^{SQ} - qV_{OC}^{rad} \right) + \left( qV_{OC}^{rad} - qV_{OC} \right)
\end{aligned}
\tag{1}
$$

$$
= \left( E_g - qV_{OC}^{SQ} \right) + q\Delta V_{OC}^{rad,\,below\,gap} + q\Delta V_{OC}^{non-rad}
\tag{2}
$$

$$
= q\Delta V_1 + q\Delta V_2 + q\Delta V_3
\tag{3}
$$

Where $V_{OC}^{SQ}$ is the maximum voltage in the SQ, and $V_{OC}^{rad}$ is the open-circuit voltage when there is only radiative recombination in the device. It is widely known that $q\Delta V_1$ is inevitable for any solar cell and relies only on the band gap ($E_g$) of the absorber for a definite solar spectrum and temperature. Here, we used the photovoltaic band gap energy ($E_g^{PV}$), which is extracted from EQE spectra to determine the $q\Delta V_1$[42] and the $E_{loss}$ (ZR1:Y6, $E_{loss} = 0.52$ eV). As shown in Table 2, ZR1:Y6 blends show the $q\Delta V_1$ value of 0.26 eV, which is similar to that of ZR1:IDIC-4Cl (0.28 eV). $q\Delta V_2$ ($q\Delta V_2 = q V_{OC}^{rad,below\,gap}$) is due to the extra radiative recombination from the non-step function absorption below the gap. Hence, Fourier-transform photo-current spectroscopy external quantum efficiency (FTPS-EQE) was applied to evaluate the $q\Delta V_2$ in the both systems. For the ZR1:IDIC-4Cl blends, due to their large HOMO offsets, a sub-gap absorption by charge-transfer (CT) states is clearly visible in the FTPS-EQE spectra (Fig. 4)[43]. While for ZR1:Y6 blends, which show reduced HOMO offsets, the FTPS-EQE

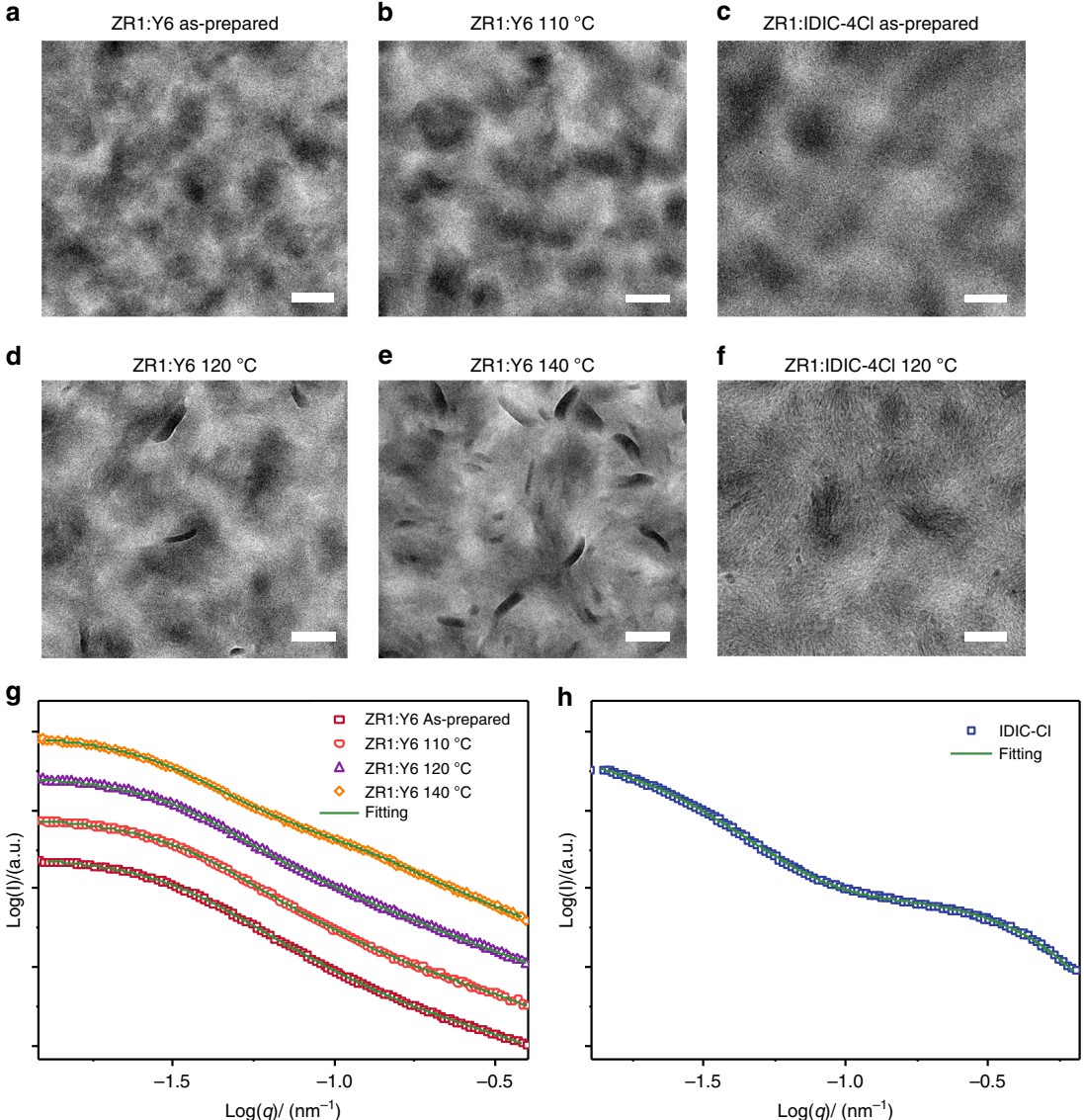

**Fig. 3** Morphology analysis of ZR1:Y6 and ZR1:IDIC-4Cl blends. **a–f** TEM images of blends films obtained at different annealing temperatures, and the scale bars are all 200 nm. **g,h** RSoXS profiles of blends films.

**Table 2 Detailed $V_{OC}$ losses of the ZR1: Y6, ZR1: IDIC-4Cl-based OPV cells.**

| Devices | $E_g$(eV) | $EQE_{EL}$ | $qV_{OC}^{SQ}$(eV) | $qV_{OC}^{rad}$(eV) | $\Delta E$(eV) | $\Delta E_1 E_{gap} - qV_{OC}^{SQ}$(eV) | $\Delta E_2\ qV_{OC}^{rad,blew\ gap}$(eV) | $\Delta E_3\ qV_{OC}^{non-rad}$(eV) | $V_{OC}^{cal}$(eV) |
|---|---|---|---|---|---|---|---|---|---|
| ZR1:Y6 | 1.38 | $1.1 \times 10^{-4}$ | 1.12 | 1.08 | 0.54 | 0.26 | 0.04 | 0.24 | 0.84 |
| ZR1:IDIC-4Cl | 1.55 | $5.0 \times 10^{-7}$ | 1.27 | 1.15 | 0.78 | 0.28 | 0.12 | 0.38 | 0.77 |

onset is almost equal to that of the Y6. The electroluminescence (EL) spectra were also in agreement with the FTPS-EQE measurements as a spectrum close to the pristine devices was observed for the ZR1:Y6 blends, while the ZR1:IDIC-4Cl blends show strong redshift as compared to the pristine devices (Supplementary Fig. 8). Hence a significant difference in the $q\Delta V_2$ values is observed for the two systems (0.04 and 0.12 eV, respectively) due to distinct and red-shifted CT absorption in the latter system. Interestingly, in contrast to the other low energy loss OSCs, which have minimized energy offsets, ZR1:Y6 blends possess relatively large energy offsets. Even though no conspicuous and red-shifted CT absorption was observed, leading to a quite small $q\Delta V_2$ of 0.04 eV[15]. The physical

mechanisms behind it are still unclear, which is out of our scope. $q\Delta V_3$ ($q\Delta V_3 = q\Delta V_{OC}^{non-rad} = -kT \ln(EQE_{EL})$) is observed due to nonradiative recombination, where $EQE_{EL}$ is radiative quantum efficiency of the photovoltaic device when charge carriers are injected in dark conditions. The enhancement of the $EQE_{EL}$ indicates reduced nonradiative recombination losses in the corresponding system. As shown in Table 2, the $EQE_{EL}$ for ZR1:Y6 ($1.1 \times 10^{-4}$) is significantly higher than ZR1:IDIC-4Cl blends ($5.0 \times 10^{-7}$), which indicates a low $q\Delta V_3$ value for ZR1:Y6 (0.24 eV) whereas, a high $q\Delta V_3$ for ZR1:IDIC-4Cl (0.38 eV) system. As a result, the ZR1:Y6 system manages to demonstrate modest energy losses and ultimately, leads to enhanced device performance under optimized hierarchical morphologies.

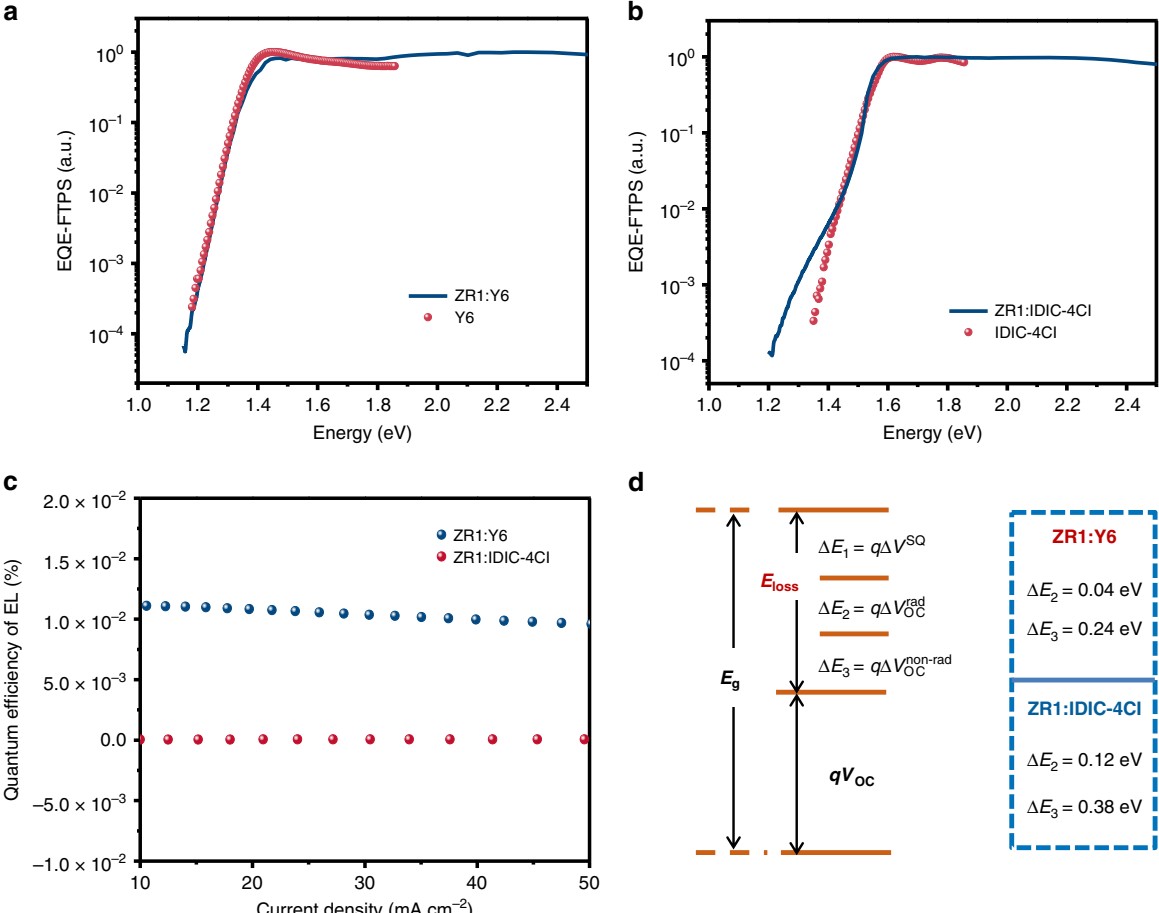

**Fig. 4** Quantification of the energy losses. **a**, **b** Fourier-transform photocurrent spectroscopy of ZR1:Y6, ZR1:IDIC-4Cl blend and corresponding single component solar cell. **c** The electroluminescence quantum efficiency of ZR1:Y6 and ZR1:IDIC-4Cl blend solar cells at different injected currents. **d** Schematic diagram for energy loss of ZR1:Y6 and ZR1:IDIC-4Cl blend solar cells.

## Discussion

Herein, we report the synthesis of a DTBDT-based small-molecule donor, ZR1, and fabricated SM-OSCs by individually blending it with Y6 and IDIC-4Cl as acceptors. The ZR1:Y6 OSCs were able to produce an excellent PCE of 14.34% (certified PCE of 14.1%), courtesy a characteristic high $J_{SC}$ of 24.34 mA cm$^{-2}$ due to a relatively broader absorption spectrum of Y6 small molecule. Furthermore, TEM and RSoXS results revealed the ZR1:Y6 blends to form an optimizing hierarchical morphology due to the high crystallinity of ZR1.The strong crystallinity of small molecules generally increase the possibility of forming oversized phase-separated domains in the blended films, leading to low $J_{SC}$ and FF values. In this study, however, the existence of hierarchical morphologies is important for the charge separation and transport, and ultimately led to a high PCE. Considering all the evaluations, especially the CCL analysis, a certain number of ZR1 crystals and amorphous ZR1-Y6 intermixed regions within large ZR1-rich domains contributed to exciton dissociation in the bulk heterojunction. This assumption is further verified by the decrease of device efficiency after being annealed at 140 °C, in which excessive ZR1 nanocrystallite aggregates led to unbalanced charge transfer with lower FF. Besides ZR1's high crystallinity that facilitates excellent charge transport, the energy loss in ZR1:Y6 has also been simultaneously reduced. A steep FTPS-EQE spectrum tail is observed in the ZR1:Y6 configuration and thus leads to a much smaller $q\Delta V_2$ (0.04 eV). Similarly, the EQE$_{EL}$ measurements for the ZR1:Y6 based device displays a high EQE$_{EL}$ of $1.1 \times 10^{-4}$, also indicating that the calculated non-radiative energy loss here is as low as 0.24 eV. All these reductions in $E_{loss}$ therefore, contribute to the increase $V_{OC}$ of the corresponding devices. Hence, this work provides opportunities to design highly efficient small-molecule donors for SM-OSCs by optimizing their hierarchical morphologies.

## Methods

**Materials**. The small-molecule donor ZR1 and the acceptor IDIC-4Cl were synthesized via referencing the reported literature. The detailed synthetic methods and characterizations of the molecular structures have been included in Supplementary Information.

**Fabrication and measurement of OSC device**. The devices were fabricated with a conventional structure of glass/ITO/PEDOT:PSS/active layer/Al. The ITO-coated glass substrates were cleaned by ultrasonic treatment in detergent, DI water, acetone and isopropyl alcohol for 20 min at each step. An interlayer of PEDOT:PSS was spin-coated at 4000 r.p.m. onto the ITO surface. The substrates were baked at 150 ° C for 15 min and then transferred into a nitrogen-filled glove box. The total concentration of mixture of ZR1 and Y6/IDIC-4Cl was ca. 15 mg ml$^{-1}$/17 mg ml$^{-1}$, and the mixture solution was stirred at 50 °C in chloroform for ca. 0.5 h until the solute was fully dissolved. Subsequently, the active layer was spin-coated from mixture solutions of ZR1+Y6 or ZR1+IDIC-4Cl. Finally, a layer of 100 nm Al layer by vacuum vapor deposition (cal. $1 \times 10^{-5}$ Pa) was used as top electrode

Newport Thermal Oriel 91159 A solar simulator was used for $J$–$V$ curves measurement under AM 1.5 G (100 mW cm$^{-2}$). Newport Oriel PN 91150 V Si-based solar cell was applied for light intensity calibration. $J$–$V$ measurement signals were recorded by a Keithley 2400 source-measure unit. Device area of each cell was approximately 4 mm$^2$. Oriel Newport system (Model 66902) equipped with a standard Si diode was used for EQEs test in air condition.

**UV visible absorption and molecular energy level measurements**. JASCO V-570 spectrophotometer was used for UV visible spectra. The energy levels were tested by the way of Cyclic voltammetry (CV) measurement which were conducted

in a 0.1 mol $L^{-4}$ tetrabutylammonium phosphorus hexafluoride (Bu4NPF6) in acetonitrile solution. Pt electrode coated with ZR1, IDIC-4Cl, Y6 films was used as working electrode, Pt plate was used as counter electrode, and Ag/Ag$^+$ electrode was used reference electrode, respectively. Redox potentials were internally calibrated using the ferrocene/ferrocenium (Fc/Fc$^+$) redox couple ($-4.8$ eV). The experiment was carried out at an electrochemical workstation (VMP3 Biologic, France).

**Charge carrier mobility, TEM, and GIWAXS characterizations**. The procedures of the device for charge carrier mobility measurement were the same for solar cell devices, except the top electrode was Au (100 nm). The current density-voltage (J–V) curves in the range of 0 to 5 V were gained by a Keithley 2400 Source-Measure Unit in the dark in the air. Tecnai G2 F20 U-TWIN TEM instrument was used for Transmission electron microscopy (TEM) test. Films for the TEM test were made by spin-coating in the same condition for solar cell devices on ITO substrates. After annealing, the blending films were immersed in water, and the floating active layers were transferred to the TEM grid.

**GIWAXS and RSoXS characterization**. GIWAXS was measured using the beamline of 7.3.3 and R-SoXS transmission was measured at a beamline of 11.0.1.2 at the Advanced Light Source (ALS). The blending or neat films for GIWAXS and R-SoXS transmission were made by the same method for device active layer except for the substrate as Si/PEDOT:PSS.

**FTPS-EQE spectra and EQE$_{EL}$**. FTPS-EQE measurement was performed at a Vertex 70 from Bruker Optics, which equipped with a quartz tungsten halogen lamp, quartz beam-splitter and external detector option. The amplification of the photocurrent product was achieved by using a low-noise current amplifier (SR570) on the illumination of the photovoltaic devices with light modulated by the Fourier transform infrared spectroscopy (FTIR). The external detector port of the FTIR gathered the signals from the current amplifier for output voltage. A Keithley 2400 SourceMeter was used for supplying voltages and recording injected current, and a Keithley 485 picoammeter was used for measuring the emitted light intensity.

## Data availability

The data supporting the results of this work are available from the corresponding author upon reasonable request. The supplementary crystallographic data for this work could be checked in The Cambridge Crystallographic Data Centre (CCDC) via www.ccdc.cam.ac.uk/data_request/cif by the series number as 1954438.

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

## Acknowledgements

The authors acknowledge the financial support from the Ministry of Science and Technology of China (Grant Nos. 2016YFA0200700, 2016YFF0203803), the National Natural Science Foundation of China (Grant No. 51961135103, 51973043, 21822503, 21534003, 21721002), the Beijing Nova Program (Grant No. Z17110001117062), Youth Innovation Promotion Association, K.C.Wong Education Foundation, and the Chinese Academy of Sciences. F.G. acknowledges the financial support from the Swedish Research Council VR (2018-06048 and 2018-05484). We acknowledge the help and suggestion from Prof. Feng Liu (Shanghai Jiao Tong University) for analyzing RSoXS data.

## Author contributions

R.Z. designed, synthesized, characterized materials, and fabricated preliminary devices. Z.J. carried out the device fabrication and characterization. C.Y. carried out the single crystal growing and transmission electron microscopy. J.Y. performed the FTPS, EL experiment was supervised by F.G. J.F, J.Z., and W.M. and they also performed GIWAXS/RSoXS measurements and data analysis. D.D. and W.Z. provided suggestion on synthesis. R.Z., Z.J., K.L., and Z.W. prepared the paper. F.G., M.A., W.M., K.L., and Z. W. helped to revise the paper. All authors discussed and commented on the paper. K.L. and Z.W. supervised the project.

## Competing interests

The authors declare no competing interests.
