## [Peer Review File · Nature Communications]

Reviewers' Comments:

Reviewer #1:

Remarks to the Author:

Developing all small molecule solar cells remain a grand challenge in the photovoltaic community. In this manuscript, Wei and co-authors reported an all small molecule organic solar cell with the highest power conversion efficiency of 14.34% while demonstrating an abnormally large domain size. The measurements of TEM and RSoXS clearly revealed the existence of large domain size larger than 130 nm. At the same time, the device exhibited a pretty low energy loss of 0.54 eV. This study was able to prove that all small molecular system can achieve a comparable efficiency with their polymer counterpart. Moreover, AFM, TEM, GIWAXS and RSoXS results on the film microstructure are quite solid, and a clear structure-property relationship has been established, which will undoubtedly help the community of OPV field to understand how to achieve such an impressive performance in all small molecule solar cells. In my opinion, the design strategy reported in this work is a significant and new advance for all small molecule solar cells, which will arouse the broad passions and interest in this class of systems. Overall, this is an interesting paper, which will come to strongly influence new materials development and processing and should be of great interest to the readers of Nature Communications as an VIP contribution. As this work is going to set a new record for all-small molecule solar cells, I am thus pleased to strongly recommend it for publication in this prestigious journal after minor revisions.

Here are some minor suggestions to improve the manuscript:

1. Typically, highly efficient organic solar cells require an optimized bulk heterojunction with phase separation on the 10–20 nm length scale. Thus the device has enough interfaces to generate the current. In the present manuscript, an abnormally large domain size of ca. 133 nm was found for all small molecule organic solar cells. How does the device generate the current efficiently?
2. In Figure 1e, the EQE of the Y6 system is much higher than that of IDIC-4Cl system in the range of 400-600 nm. The author should give some comments on this point.
3. The DSC of the pure molecules and blend materials should be measured for a deeper analysis of the abnormal phase separation behavior (for instance, miscibility).

Reviewer #2:

Remarks to the Author:

In this paper, Zhou et al. reported two highly efficient all-small-molecule organic solar cells (ZR1:IDIC-4Cl and ZR1:Y6) with efficiencies of 9.64% and 14.34%, respectively. This performance is quite impressive, and the strategies of designing small molecule donors and acceptors were instructive. The author applied X-ray single crystal structural analyses, TEM, GIWAXS and RSoXS to clearly characterize the packing mode and domain size of two kinds of SM-based active layers. Interestingly, they found that the large domain sizes (higher than 100 nm) also can demonstrate sufficient D:A interfaces for effective exciton dissociation with low charge recombination, which is contrary to our previous understanding of bulk heterojunctions. Also, from FTPS-EQE, they found ZR1:Y6 blends possess relatively large energy offsets but no un conspicuous and red-shifted CT absorption was observed. All these attractive phenomena will receive the attention of the broad community engaged in this field, therefore, I recommend for publication in Nature Communications after addressing the following comments:

1. In this work, the author chose bithiophene as the n-bridge for novel small molecule donor ZR1 with an A-n-D-n-A architecture. So the readers might wonder how much difference it would make to replace the bithiophene with trithiophene, have the author performed any comparative experiments?

2. From the CV measurement, the author found that the LUMO energy levels of two acceptors (IDIC-4Cl and Y6) were calculated to be at the same value (-4.10 eV), but the Vocs of devices based on them were significantly different, the reason should be explained in the main text.

3. In this paper, the author use thermal annealing to enhance the domain size, could solvent vapor annealing reach the same or even better result?

4. The author stated that the presence of a small number of ZR1 crystals are indicative of facilitating the charge transport, while excessive crystals would lead to a reduction in the charge generation (exciton separation) efficiency. Also the Jsc of this system seems quite high. While the authors attributed it to the non-radiative energy loss of ZR1:Y6 with a high EQEEL of 1.1×10^{-4} , further intrinsic reasons/mechanism deem to be well explained.

5. Also there are some mistakes should be corrected.

1) In Page 9, line 182 and 183, there should be spaces between numbers and units "nm".

2) In the text, there should be no spaces between numbers and units "°C".

3) In Page 11, line 228, " $[qV]_{oc}^{sq}$ " should be " V_{oc}^{sq} ".

6. The term "novel" was mentioned many times in description of the molecule ZR1. In fact, there is rather similarity between the structures reported earlier such as (Sci China Chem, 2017, 60, 552), and furthermore the refs are not even cited.

7. Lastly, since it is a record for all small mol based OPV, I would strongly suggest to have a third party certification to support.

Reviewer #3:

Remarks to the Author:

The manuscript provides a narrative describing small-molecule OPV solar cells. The abstract claims a high power conversion efficiency of 14.34%. The rest of the manuscript focuses on morphology characterization with claims of an "abnormal" domain size of 130 nm. Although the efficiency is impressive, the morphology analysis is not conclusive, there are some parts of it performed incorrectly, and even if the domain sizes prove to be of the correct magnitude, the novelty of the results is grossly exaggerated – these morphology traits are not at all uncommon in OPVs. The paper should not be published in its current form in any journal.

If the morphology analysis were fixed and properly contextualized, the paper would be suitable for publication in some journal, but I am not sure that it meets the bar for potential impact that is expected of a Nature family journal. From a device engineering perspective, it is notable. It is an incremental increase in reported all-small-molecule photovoltaic efficiency (from the previous 13.6%), it was achieved with a binary system, and furthermore it is especially impressive that the average of 10 devices is 14.2%. Such reproducibility across multiple devices is unusual.

For comparison, however, a very similar recent paper in a Nature family journal, Zhou et al., Nature Energy 3, pages 952–959 (2018), reported a similar increase in efficiency from previous studies, but also included a detailed study of binary vs. ternary behavior, a more complete morphological workup, measurements of charge generation, extraction and recombination leading to optimized carrier dynamics, and a discussion of improving Voc. The manuscript currently under consideration does not have such a broad scope with general conclusions that might serve as a guide for others not working with these specific systems. Thus, the manuscript under consideration might be a better fit for a journal such as the excellent J Mater Chem A, which has recently been found to be a suitable journal for reporting all-small-molecule OPV record efficiencies such as in X Li et al., J. Mater. Chem. A 2019, 7, 3682, a paper with which the manuscript under consideration has many similarities.

My specific concerns about the morphology analysis are listed below.

- The quantitative "domain sizes" are extracted from the RSoXS, which is analyzed incorrectly. Most importantly, the length scales are extracted from Iq^2 vs. q representation, which is incorrect. It is only appropriate to extract domain sizes from a I vs. q representation. I vs. q is, for example, what appears to have been done for the RSoXS in Zhou et al., Nature Energy 3, pages 952–959 (2018). Although many RSoXS papers use Iq^2 vs. q for domain sizes, it is not correct. This is covered in Ye, Stuard, and Ade's Chapter of "Conjugated Polymers: Properties, Processing, and Applications, CRC Press, 2019, where Ade dryly notes that this issue "is still debated indirectly in the R-SoXS community." In this same chapter Ade also presents Figure 13.4 showing similar length scales being extracted from I vs. q and Iq^2 vs q representations, which I appreciate, but this is not the case for all scattering curves. If the authors in the manuscript currently under review can show that the same length scales can be credibly extracted from an I vs. q representation, I would accept these length scales as representative of morphological traits in the measured films.

- Even if the larger length scales of compositional fluctuation are correct, there is nothing particularly "abnormal" and "contrary to our previous understanding of bulk heterojunctions" [line 214] about these larger domain sizes unless they are pure. If the domains were large and pure, it would indicate an unusual exciton diffusion length or some other emergent phenomenon. But they are not shown to be pure, and in fact these 'domains' are likely of mixed composition. Only the narrow crystals at 120°C and 140°C are nominally pure phases. Although we have no other measurement of absolute phase purity in this manuscript, in their diffraction discussion the authors find coherence lengths <10nm, and discuss the need for phase mixing within the larger "domains" to ensure charge transport. This morphology is thus no different than the dozens of published reports of "hierarchical" morphology traits in OPVs, where larger-scale composition fluctuations occur over 100s of nanometers with more finely divided, compositionally mixed structures within these fluctuations on 10s of nanometers or smaller scales. See, for example, Wei Chen's "Hierarchical Nanomorphologies Promote Exciton Dissociation in Polymer/Fullerene Bulk Heterojunction Solar Cells." Nano Lett. 2011, 11, 3707, wherein larger scale composition fluctuations of 100-300 nm are observed (very similar to manuscript under review!), with finely divided mixed phases inside each larger-scale domain. J Wen et al., "Visualization of Hierarchical Nanodomains in Polymer/Fullerene Bulk Heterojunction Solar Cells." Microscopy and Microanalysis 2014, 20, 1507 gives a similar account with similar length scales of large-scale compositional heterogeneity. There are many more publications on hierarchical morphologies in OPV that I could cite, and many more details from the paper under review that indicate a 'classic' hierarchical morphology without large pure domains, but I trust my point is made. Given all these considerations, the morphology analysis should be recontextualized, the language about the 'unusual' nature of the morphology should all be removed, and the similarity to other hierarchical morphologies common to OPV should instead be discussed.

Response to the comments of reviewer

We would like to thank the referees for spending time on this paper and providing invaluable comments which substantially helped to improve the quality of the paper. The manuscript has been carefully revised according to the comments.

Reviewer #1 (Remarks to the Author):

General comments: Developing all small molecule solar cells remain a grand challenge in the photovoltaic community. In this manuscript, Wei and co-authors reported an all small molecule organic solar cell with the highest power conversion efficiency of 14.34% while demonstrating an abnormally large domain size. The measurements of TEM and RSoXS clearly revealed the existence of large domain size larger than 130 nm. At the same time, the device exhibited a pretty low energy loss of 0.54 eV. This study was able to prove that all small molecular system can achieve a comparable efficiency with their polymer counterpart. Moreover, AFM, TEM, GIWAXS and RSoXS results on the film microstructure are quite solid, and a clear structure-property relationship has been established, which will undoubtedly help the community of OPV field to understand how to achieve such an impressive performance in all small molecule solar cells. In my opinion, the design strategy reported in this work is a significant and new advance for all small molecule solar cells, which will arouse the broad passions and interest in this class of systems. Overall, this is an interesting paper, which will come to strongly influence new materials development and processing and should be of great interest to the readers of Nature Communications as an VIP contribution. As this work is going to set a new record for all-small molecule solar cells, I am thus pleased to strongly recommend it for publication in this prestigious journal after minor revisions.

Author response: We thank the reviewer for positive comments to this manuscript.

Comment 1. Typically, highly efficient organic solar cells require an optimized bulk heterojunction with phase separation on the 10–20 nm length scale. Thus the device has enough interfaces to generate the current. In the present manuscript, an abnormally large domain size of ca. 133 nm was found for all small molecule organic solar cells. How does the device generate the current efficiently?

Author response: Thank you for your comment. The efficient current generation has been the most

important point for the revision. For our system, the large donor phase is responsible for the efficient charge transport, whereas the smaller donor phase accounts for efficient charge separation. Therefore, instead of emphasizing the large domain size, we are focusing more on the hierarchical phase separation. In the hierarchical morphology of all small molecular blends, the large donor or acceptor rich domains (ca. 70 nm), as well as the donor-acceptor blend phase (ca. 10 nm), coexist within the same domains. The revised explanation has been changed in the revision accordingly.

Comment 2. In Figure 1e, the EQE of the Y6 system is much higher than that of IDIC-4Cl system in the range of 400-600 nm. The author should give some comments on this point.

Author response: Thanks for the reviewer's comment. As mentioned in the manuscript, the two systems have similar D/A ratios as well as FFs. Therefore, the EQE difference should originate from a more efficient charge separation in the ZR1:Y6 system. Although both systems show a hierarchical phase separation, as measured by RSoXs, their TEM images are quite different. The donor-acceptor blend phase within the large domains was not observed by TEM in ZR1:Y6 blends, while a clear phase separation was detected in ZR1:IDIC-4Cl system. This difference in morphology is indicative of good miscibility between ZR1 and Y6, which ensures a large D-A interface for efficient charge separation.

Comment 3. The DSC of the pure molecules and blend materials should be measured for a deeper analysis of the abnormal phase separation behavior (for instance, miscibility).

Author response: Thanks for the reviewer's comment. As per your instructions, the DSC of the pure molecules and the blending materials have been measured for the revision. The details from the DSC test, however, were not so contributive for explaining the abnormal phase separation behavior. By comparison, the peak from ZR1:IDIC-4Cl blend had a more obvious shift from the pure ZR1 as compared to ZR1:Y6 system. Likewise, only one peak appeared in the ZR1:Y6 blend, whereas, there were two peaks in the ZR1: IDIC-4Cl system, the first one originating from the IDIC-4Cl. To sum up the above-mentioned, the ZR1:Y6 system appears to have a better miscibility than ZR1:IDIC-4Cl system, which can help in explaining the difference in the device performance of the two systems.

Figure R1. The DSC heat-only thermograms of pure small molecules and their blends.

Reviewer #2 (Remarks to the Author):

General comments: In this paper, Zhou et al. reported two highly efficient all-small-molecule organic solar cells (ZR1:IDIC-4Cl and ZR1:Y6) with efficiencies of 9.64% and 14.34%, respectively. This performance is quite impressive, and the strategies of designing small molecule donors and acceptors were instructive. The author applied X-ray single crystal structural analyses, TEM, GIWAXS and RSoXS to clearly characterize the packing mode and domain size of two kinds of SM-based active layers. Interestingly, they found that the large domain sizes (higher than 100 nm) also can demonstrate sufficient D:A interfaces for effective exciton dissociation with low charge recombination, which is contrary to our previous understanding of bulk heterojunctions. Also, from FTPS-EQE, they found ZR1:Y6 blends possess relatively large energy offsets but no un conspicuous and red-shifted CT absorption was observed. All these attractive phenomena will receive the attention of the broad community engaged in this field, therefore, I recommend for publication in Nature Communications after addressing the following comments.

Author response: We thank the reviewer for positive comments to this manuscript.

Comment 1: In this work, the author chose bithiophene as the π -bridge for novel small molecule donor ZR1 with an A- π -D- π -A architecture. So the readers might wonder how much difference it would make to replace the bithiophene with trithiophene, have the author performed any comparative experiments?

Author response: The comparative experiments are important for novel molecules. We have synthesized three molecules with monothiophene, bithiophene, trithiophene as π bridges, namely ZR1-T, ZR1, ZR1-3T, respectively. The molecule structure, UV-absorption spectrum, molecule energy levels and device performance are shown as following (Figure R2, R3 and Table R1, respectively):

Figure R2. ZR1, ZR1-T, ZR1-3T molecular structures.

Figure R3. a) Normalized UV–vis absorption spectra of ZR1, ZR1-T, ZR1-3T in solution and thin films. b) Energy diagrams of ZR1, ZR1-T, ZR1-3T.

Table R1: Device comparison of ZR1:Y6, ZR1-T:Y6, ZR1-3T:Y6 blends at TA 120°C.

Donor/Acceptor	D:A	V_{OC} [V]	J_{SC} [mA cm^{-2}]	FF [%]	PCE [%]
ZR1:Y6	1:0.5	0.861	24.34	68.44	14.34
ZR1-T:Y6	1:0.6	0.498	0.0019	25.29	0.025
ZR1-3T:Y6	1:0.5	0.754	22.51	45.47	7.67

As we can see from the data, the increase in the π bridge length leads to a redshift in the absorption spectra, as well as upshifts of the HOMO levels. The devices based on ZR1-T:Y6 blends exhibited extremely poor performance as the HOMO of ZR1-T and Y6 lie at almost the same energy levels, and hence can't provide enough driving force for exciton dissociation and charge transport. For ZR1-3T blend, a relatively higher HOMO level led to a low V_{OC} of 0.754V as compared to 0.861V for the ZR1 system. Likewise, the J_{SC} and FF of ZR-3T: Y6 blends were also lower than ZR1-3T: Y6 blends, resulting in a much lower efficiency than ZR1

The modification on Page 8 is as follows:

Two donor small molecules with monothiophene and trithiophene as π bridges, namely ZR1-T and ZR1-3T, respectively, were also synthesized for the sake of comparison. The molecule structure, UV-absorption spectrum, molecule energy levels and device performance are exhibited in Supplementary Fig. 3, 4, and Supplementary Table 6, respectively. The devices based on ZR1-T:Y6 blends exhibited extremely poor performance as the HOMO of ZR1-T and Y6 lie at almost the same energy levels, and hence can't provide enough driving force for exciton dissociation and charge transport. For ZR1-3T blend, a relatively higher HOMO level led to a low V_{OC} of 0.754V as compared to 0.861V for the ZR1 system. Likewise, the J_{SC} and FF of ZR-3T: Y6 blends were also lower than ZR1-3T: Y6 blends, resulting in much lower efficiency than ZR1.

Comment 2. From the CV measurement, the author found that the LUMO energy levels of two acceptors (IDIC-4Cl and Y6) were calculated to be at the same value (-4.10 eV), but the V_{oc} 's of

devices based on them were significantly different, the reason should be explained in the main text.

Author response: Thanks for the reviewer's comment. While conducting the energy loss analysis (page 12), we explained the reasons leading to the difference in V_{oc} . A steep FTPS-EQE spectrum tail is observed in the ZR1:Y6 configuration and thus leads to a much smaller $q\Delta V_2$ (0.04 eV). Similarly, the EQEEL measurements for the ZR1:Y6 based devices display a high EQE_{EL} of 1.1×10^{-4} , further indicating that the calculated non-radiative energy loss here is as low as 0.24 eV. All these reductions in E_{loss} therefore, contribute to the increase V_{oc} of the ZR1:Y6 devices

Comment 3: In this paper, the author use thermal annealing to enhance the domain size, could solvent vapor annealing reach the same or even better result?

Author response: Thanks for the reviewer's comment. While device optimization, we did try the solvent vapor annealing along with the thermal annealing for both systems. The solvent vapor annealing, however, didn't show a positive impact as compared to the solely thermal annealed devices, as a decrease in the J_{sc} and FF was observed when the corresponding devices were solvent vapor annealed with THF. (Table R2)

Table R2: Device optimization of solvent annealing with THF for ZR1: Y6 at TA 120°C

Donor/ Acceptor	D:A	V_{oc} [V]	J_{sc} [mA cm^{-2}]	FF [%]	TA (°C)	SVA	PCE [%]
ZR1:Y6	1:0.5	0.861	24.34	68.44	120	no	14.34
ZR1:Y6	1:0.6	0.851	22.46	62.58	120	THF (40s)	11.97

The modification has been made on Page 7 as following:

The solvent vapor annealing, however, didn't show a positive impact as compared to the solely thermal annealed devices, as a decrease in the J_{sc} and FF was observed when the corresponding devices were solvent vapor annealed with THF (Supplementary Table 5).

Comment 4: The author stated that the presence of a small number of ZR1 crystals are indicative of facilitating the charge transport, while excessive crystals would lead to a reduction in the charge generation (exciton separation) efficiency. Also the J_{sc} of this system seems quite high. While the

authors attributed it to the non-radiative energy loss of ZR1:Y6 with a high EQE_{EL} of 1.1×10^{-4} , further intrinsic reasons/mechanism deem to be well explained.

Author response: Thank you for your comments. In the corresponding devices, the existence of hierarchical morphologies indicated that it is the smaller blend phase that can be accounted for efficient charge separation, whereas the larger donor phase is responsible for charge transport. Both these attributes consequently contributed to the photocurrent.

Considering the thermal annealing conditions, the devices annealed at 120°C and 140°C led to a high-power conversion efficiency, with the former being a fraction better. A closer look revealed that for the devices annealed at 120°C , an optimum number of ZR1 nanocrystallites, as well as amorphous ZR1-Y6 intermixed regions within large ZR1 rich domains were present to achieve efficient charge separation and charge transport in the heterojunction. Thermal annealing (TA) at 140°C , on the other hand, led to the formation of relatively excessive ZR1 nanocrystallite aggregates as compared to the former system. Hence, even though the amorphous ZR1-Y6 intermixed regions within the devices annealed at 140°C can also reach effective exciton dissociation and charge transport and thus high J_{SC} , the excessive ZR1 nanocrystallite aggregate formation might lead to a relatively rough active layer surface, resulting in a reduced FF, and ultimately a lower PCE.

Considering the intrinsic reasoning of a high EQE_{EL} , since EQE_{EL} can be related to the low non-radiative energy loss as:

$$q\Delta V_3 = q\Delta V_{oc}^{nonrad} = -kT \ln(EQE_{EL})$$

An increase of EQE_{EL} will lead to a lower ΔV_{oc}^{nonrad} , which has been mentioned in the manuscript.

Comment 5: Also there are some mistakes should be corrected.

- 1) In Page 9, line 182 and 183, there should be spaces between numbers and units "nm".
- 2) In the text, there should be no spaces between numbers and units " $^\circ\text{C}$ ".
- 3) In Page 11, line 228, " $[\text{qV}]_{oc}^{sq}$ " should be " V_{oc}^{sq} ".

Author response: Thanks for the reviewer's detailed comments. These typo mistakes have been corrected in the revision.

The modification on Page 10 has been made as follows:

Likewise, the CCL (100) of the ZR1:Y6 film in the OOP direction turned out to be 4.07 nm, 6.21 nm,

6.50 nm, 6.90 nm for the as-cast and films annealed at 110°C, 120°C and 140°C, respectively.

The modification on Page 12 is as follows:

Where V_{OC}^{SQ} is the maximum voltage in the SQ, and V_{OC}^{rad} is the open-circuit voltage when there is only radiative recombination in the device.

Comment 6: The term “novel” was mentioned many times in description of the molecule ZR1. In fact, there is rather similarity between the structures reported earlier such as (Sci China Chem, 2017, 60, 552), and furthermore the refs are not even cited.

Author response: Thanks for the reviewer’s comment. Although there is a similarity in the structure between ZR1 and that in literature, the type and length of side chain on center core, π bridge, terminal group are different. Those factors have an obvious impact on molecular properties and device performance according the previous report such as (*J. Am. Chem. Soc.* **139**, 5085-5094 (2017); *Adv. Energy Mater.* **9**, 1803175 (2019); *Chem. Mater.* **30**, 2129-2134 (2018)). We have revised reference in the revised manuscript, and remove most of the word “novel” in the revision.

Comment 7: Lastly, since it is a record for all small molecule based OPV, I would strongly suggest to have a third party certification to support.

Author response: Thanks for the reviewer’s comment. A third-party certification has been carried out, which gave an efficiency of 14.1 % with J_{SC} of 24.00 mAcm⁻², a V_{OC} of 0.836V and a fill factor (FF) of 70.2% (Supplementary Fig2). The certification confirmed that all-small-molecule organic solar cell devices can indeed endure the standard measurement and achieve results relatively close to our regular testing, indicating a reliable and reproducible high performance of our systems.

The modification has been made on Page 6 as:

The device based on ZR1: Y6 was certified at an accredited laboratory, certifying a PCE of 14.1% (Supplementary Fig2). Notably, all devices were fabricated without any additive and electron-transporting layer, making them an important prospect for future industrial manufacturing.

Reviewer #3 (Remarks to the Author):

General comment: The manuscript provides a narrative describing small-molecule OPV solar cells. The abstract claims a high power conversion efficiency of 14.34%. The rest of the manuscript focuses on morphology characterization with claims of an “abnormal” domain size of 130 nm. Although the efficiency is impressive, the morphology analysis is not conclusive, there are some parts of it performed incorrectly, and even if the domain sizes prove to be of the correct magnitude, the novelty of the results is grossly exaggerated – these morphology traits are not at all uncommon in OPVs. The paper should not be published in its current form in any journal. If the morphology analysis were fixed and properly contextualized, the paper would be suitable for publication in some journal, but I am not sure that it meets the bar for potential impact that is expected of a Nature family journal. From a device engineering perspective, it is notable. It is an incremental increase in reported all-small-molecule photovoltaic efficiency (from the previous 13.6%), it was achieved with a binary system, and furthermore it is especially impressive that the average of 10 devices is 14.2%. Such reproducibility across multiple devices is unusual. For comparison, however, a very similar recent paper in a Nature family journal, Zhou et al., Nature Energy 3, pages 952–959 (2018), reported a similar increase in efficiency from previous studies, but also included a detailed study of binary vs. ternary behavior, a more complete morphological workup, measurements of charge generation, extraction and recombination leading to optimized carrier dynamics, and a discussion of improving Voc. The manuscript currently under consideration does not have such a broad scope with general conclusions that might serve as a guide for others not working with these specific systems. Thus, the manuscript under consideration might be a better fit for a journal such as the excellent J Mater Chem A, which has recently been found to be a suitable journal for reporting all-small-molecule OPV record efficiencies such as in X Li et al., J. Mater. Chem. A 2019, 7, 3682, a paper with which the manuscript under consideration has many similarities.

Author response: Thanks for the reviewer’s detailed comments.

For the morphology, we have considered the reviewer’s comments carefully and realized that in the previous version, the large domain size of ca. 130 nm is over-emphasized. Although large domains in TEM images were observed and were further confirmed by RSoXS analysis, the smaller domains also contributed immensely for improving device performance. Therefore, the title of the manuscript has been changed to “All Small Molecule Organic Solar Cells with Over 14% Power Conversion Efficiency by Optimizing Hierarchical Morphologies”, and the manuscript has been revised carefully.

Especially, we have analyzed the RSoXS and the TEM results more carefully, and have identified the hierarchical morphologies with narrow crystals. Since the formation of large crystals leads to a reduction in the efficiency for most of the reported systems, like in Joule 3, pages 1-15 (2019), achieving a boosted efficiency due to the presence of a small number of crystals, such as in our systems, is quite astonishing.

Data reproducibility: In order to evaluate the reproducibility of the best results, we prepared three glass substrates, where each substrate can bear four solar cells. Since spin-coating methodology was employed to solution process the corresponding devices (low speeds and small amounts of solution), occasionally, a non-uniform deposition occurred over the glass substrate such that the active layer solution was unable to uniformly coat at least one of the four cells. The device performance from such cells (only the uncovered part of the substrate) would be low or almost zero, and hence would not be meaningful for statistical significance. However, as far as the comparison between the glass substrates is concerned, the device performances were similar to each other. Therefore, we calculated the average device performance from the three glass substrates (12 devices, 2 uncovered) fabricated under the best device conditions. Thus, to get a conclusive reproducibility data, we fabricated two different batches of about 10 and 20 devices each. Their average device performance has been tabulated in Table R3 and R4, respectively, which indicated a reliable and reproducible high performance of our systems.

Table R3: The average device performance of 10 cells.

Number	Donor/ Acceptor	V_{OC} [V]	J_{sc} [mA cm ⁻²]	FF [%]	PCE [%]
1	ZR1:Y6	0.860	24.40	67.63	14.20
2	ZR1:Y6	0.858	24.57	67.50	14.24
3	ZR1:Y6	0.852	25.07	66.72	14.25
4	ZR1:Y6	0.857	24.65	67.57	14.28
5	ZR1:Y6	0.858	24.52	67.92	14.28
6	ZR1:Y6	0.860	24.62	67.56	14.30
7	ZR1:Y6	0.858	24.77	67.26	14.30
8	ZR1:Y6	0.855	25.07	66.83	14.33
9	ZR1:Y6	0.854	25.09	66.84	14.33
10	ZR1:Y6	0.861	24.34	68.44	14.34
Average		0.857	24.75	67.40	14.27

Table R4: The average device performance of 10 cells

Number	Donor/ Acceptor	V _{OC} [V]	J _{SC} [mA cm ⁻²]	FF [%]	PCE [%]
1	ZR1:Y6	0.856	24.33	67.45	14.04
2	ZR1:Y6	0.865	24.70	65.88	14.07
3	ZR1:Y6	0.856	24.39	67.74	14.14
4	ZR1:Y6	0.855	25.18	65.74	14.15
5	ZR1:Y6	0.855	24.77	66.92	14.16
6	ZR1:Y6	0.864	24.65	66.49	14.16
7	ZR1:Y6	0.865	23.97	68.27	14.16
8	ZR1:Y6	0.860	24.86	66.28	14.17
9	ZR1:Y6	0.860	24.40	67.63	14.20
10	ZR1:Y6	0.858	24.60	67.31	14.20
11	ZR1:Y6	0.859	24.55	67.43	14.22
12	ZR1:Y6	0.858	24.57	67.50	14.24
13	ZR1:Y6	0.852	25.07	66.72	14.25
14	ZR1:Y6	0.857	24.65	67.57	14.28
15	ZR1:Y6	0.858	24.52	67.92	14.28
16	ZR1:Y6	0.860	24.62	67.56	14.30
17	ZR1:Y6	0.858	24.77	67.26	14.30
18	ZR1:Y6	0.855	25.07	66.83	14.33
19	ZR1:Y6	0.854	25.09	66.84	14.33
20	ZR1:Y6	0.861	24.34	68.44	14.34
Average		0.858	24.66	67.19	14.22

In terms of comparison with the recent paper by Zhou et al., Nature Energy 3, pages 952–959 (2018), although there is a formation of the hierarchical morphology in the ternary BHJ blend via appropriate material selection, while identifying large-scale phase separation (PC₇₁BM to the non-fullerene mixture) from the small-scale phase separation in the refined non-fullerene network, the poor morphology due to a good BTR:NITI mixing led to a strong charge recombination and thus, a poor FF (46.99±2.98%) and ultimately a low PCE of 6.82% for the binary blends. In our manuscript, binary all-small molecules organic solar cells can achieve the highest power conversion efficiency

(PCE) of 14.34 % with optimizing hierarchical nanomorphologies, in which the large donor or acceptor rich domains (ca. 70 nm), as well as the donor-acceptor blend phase (ca. 10 nm), and even the donor crystals of tens of nanometers coexist within the same domains.

For comparison with the paper in *J. Mater. Chem. A* 2019, 7, 3682, we present a broad scope with general conclusions that might serve as a guide for others not working with these specific systems. Such as 1) We fabricated binary all small molecule organic solar cells with hierarchical nanomorphologies capable of producing high PCEs of more than 14%. 2) By regulating the crystallinity and thus controlling the morphology, we have designed and succeeded in getting several systems with more than 14% PCE along this line of thought, which is of great significance for the design of high-efficiency donor materials for organic solar cells. 3) Compared with the PM6:Y6 system, although the HOMO of ZR1 is much higher than PM6, their V_{oc} is similar, which provides a new idea for the design of better OPV donor materials.

Comment 2: The quantitative “domain sizes” are extracted from the RSoXS, which is analyzed incorrectly. Most importantly, the length scales are extracted from Iq^2 vs. q representation, which is incorrect. It is only appropriate to extract domain sizes from a I vs. q representation. I vs. q is, for example, what appears to have been done for the RSoXS in Zhou et al., *Nature Energy* 3, pages 952–959 (2018). Although many RSoXS papers use Iq^2 vs. q for domain sizes, it is not correct. This is covered in Ye, Stuard, and Ade’s Chapter of “Conjugated Polymers: Properties, Processing, and Applications, CRC Press, 2019, where Ade dryly notes that this issue “is still debated indirectly in the R-SoXS community.” In this same chapter Ade also presents Figure 13.4 showing similar length scales being extracted from I vs. q and Iq^2 vs q representations, which I appreciate, but this is not the case for all scattering curves. If the authors in the manuscript currently under review can show that the same length scales can be credibly extracted from an I vs. q representation, I would accept these length scales as representative of morphological traits in the measured films.

Author response: Thank you for your detailed comments. We acknowledge that using Iq^2 vs. q plot to get domain sizes is still debated among the scientific community. But since we did find large domains from TEM images, we went on with Iq^2 vs. q for domain size measurements as there are many published results that employ RSoXS in this manner. Therefore, to get quantified domain sizes we plotted RSoXS data as Iq^2 vs. q . As you have pointed out, this method would not be correct.

Hence, in the revised manuscript we extracted the domain sizes from the I vs. q representation as the reviewer suggested. The RSoXS data were fitted by a unified model to get quantified domain sizes. Large domains with ca. 70 nm and small domains with ca. 10 nm are found. We have corrected the manuscript with the new results.

3. - Even if the larger length scales of compositional fluctuation are correct, there is nothing particularly “abnormal” and “contrary to our previous understanding of bulk heterojunctions” [line 214] about these larger domain sizes unless they are pure. If the domains were large and pure, it would indicate an unusual exciton diffusion length or some other emergent phenomenon. But they are not shown to be pure, and in fact these ‘domains’ are likely of mixed composition. Only the narrow crystals at 120°C and 140°C are nominally pure phases. Although we have no other measurement of absolute phase purity in this manuscript, in their diffraction discussion the authors find coherence lengths <10nm, and discuss the need for phase mixing within the larger “domains” to ensure charge transport. This morphology is thus no different than the dozens of published reports of “hierarchical” morphology traits in OPVs, where larger-scale composition fluctuations occur over 100s of nanometers with more finely divided, compositionally mixed structures within these fluctuations on 10s of nanometers or smaller scales. See, for example, Wei Chen’s “Hierarchical Nanomorphologies Promote Exciton Dissociation in Polymer/Fullerene Bulk Heterojunction Solar Cells.” *Nano Lett.* 2011, 11, 3707, wherein larger scale composition fluctuations of 100-300 nm are observed (very similar to manuscript under review!), with finely divided mixed phases inside each larger-scale domain. J Wen et al., “Visualization of Hierarchical Nanodomains in Polymer/Fullerene Bulk Heterojunction Solar Cells.” *Microscopy and Microanalysis* 2014, 20, 1507 gives a similar account with similar length scales of large-scale compositional heterogeneity. There are many more publications on hierarchical morphologies in OPV that I could cite, and many more details from the paper under review that indicate a ‘classic’ hierarchical morphology without large pure domains, but I trust my point is made.

Given all these considerations, the morphology analysis should be recontextualized, the language about the ‘unusual’ nature of the morphology should all be removed, and the similarity to other hierarchical morphologies common to OPV should instead be discussed.

Author response: We deeply thank the referee's helpful comments. Firstly, we earnestly accept your

considerations about the morphology analysis. However, we really cherish this opportunity to set forth the progress and merits of this article and hope our revisions could be conclusive and novel. The article has been seriously revised to thoroughly explain the morphology of the active layers.

The morphology analysis has been recontextualized, the language about the ‘unusual’ nature of the morphology has all been removed, and the hierarchical morphology analysis has been added. For example, we changed the description from “we achieved the highest power conversion efficiency (PCE) of 14.34 % for binary all small molecule organic solar cells while demonstrating an abnormally large domain size of ca. 133 nm” to “we achieved the highest power conversion efficiency (PCE) of 14.34 % for binary all small molecule organic solar cells by optimizing the hierarchical nanomorphologies, in which the large donor or acceptor rich domains (ca. 70 nm), as well as the donor-acceptor blend phase (ca. 10 nm), and even the donor crystals of tens of nanometers coexist within the same domains.”(Page 2);

For RSoXS analysis (Page 11), the modification is “These results suggest the formation of a hierarchical morphology within the blend i.e. a smaller donor phase, having a size very close to the exciton diffusion length of ca. 10 nm and accounting for the efficient charge separation, whereas a larger donor phase that is responsible for the efficient charge transport within the system. In contrast to the previous reports about the SM-OSCs’ hierarchical morphology, the donor crystals, about 100 nm long and 30 nm wide, were also observed. All these phenomena consequently contribute to the effective photocurrent, indicating that the nano-structural characteristics with multiple length scales as well as coexisting crystals are key factors for high performance.”.

In the discussion part (Page 13 and 14), the modification is “TEM and RSoXS results revealed the ZR1:Y6 blends to form optimizing hierarchical nanomorphology due to the high crystallinity of ZR1. Small molecules generally tend to crystallize and easily form oversized phase-separated domains in the blended films, leading to low JSC and FF values. In this study, however, the existence of hierarchical morphologies balanced the charge separation and charge transport, and therefore high PCE was obtained. Considering all the evaluations, especially the CCL analysis, a certain number of ZR1 crystals and amorphous ZR1 – Y6 intermixed regions within large ZR1-rich domains contributed to exciton dissociation in the bulk heterojunction. This assumption is further verified by the decrease of device efficiency after being annealed at 140° C, in which excessive ZR1 nanocrystallite aggregates led to unbalanced charge transfer with lower FF.”.

One of the disadvantages of all small molecular OSCs, as compared with PSCs, is the difficulty in forming an effective interpenetrating network structure. In this article, ZR1 molecule showed strong crystallinity and compact molecular packing, while blending it with Y6 indicated an optimized hierarchical morphology, forming an ideal state where the large domain (ca. 70 nm), the common domain (ca. 10 nm) as well as the pure crystals co-existed together, and all had an efficient cooperation in current generation. This design strategy can, therefore, provide an efficient way to enhance charge mobility and ultimately the PCEs of all small molecular OSCs. Another advantage of ZR1 is that the optimized ZR1:Y6 based devices exhibited a low E_{loss} of 0.52 eV, which is uncommon in both, all small molecular OSCs and PSCs. The energy loss analysis conducted for these two systems would be helpful for further research.

Reviewers' Comments:

Reviewer #1:

Remarks to the Author:

The manuscript "All Small Molecule Organic Solar Cells with Over 14% Power Conversion Efficiency by Optimizing Hierarchical Morphologies" by Prof. Zhixiang Wei and coworkers has undergone substantial revisions, which I believe have clarified most of the questions raised in the course of its review.

To address the major concerns of three reviewers, they made the following the key improvements:

- 1) Modified the title used in the paper: A more appropriate title "All Small Molecule Organic Solar Cells with Over 14% Power Conversion Efficiency by Optimizing Hierarchical Morphologies" was used.
- 2) Explained the EQE difference in the range of 400-600 nm.
- 3) Included a third party certification for confirming the high efficiency.
- 4) Performed DSC analysis of the pure small molecules and their blends.
- 5) Analyzed and discussed the RSoXS and the TEM data more carefully, and importantly, identified the hierarchical morphologies with narrow crystals.

Overall, these revisions provided a clear mechanistic explanation of the high performance all small molecule solar cells. I am convinced that the authors have made substantial changes to address the concerns of all reviewers and the data are now supportive of the conclusion. I am thus happy to recommend it for publication in Nature Communications without further changes.

Reviewer #2:

Remarks to the Author:

I am OK with the revision and it could be accepted now.

Reviewer #3:

Remarks to the Author:

The authors have re-characterized the morphology discussion entirely, embracing my suggestion that the observed morphology should be classified as hierarchical, and also adopting most of my other suggestions. I am satisfied that my previous round of concerns have been mostly addressed. Half of the paper is rewritten, and there is a lot of new prose to evaluate.

My principal concern about the new prose is that it is too strong in its assertions about the role of the hierarchical morphology in producing the excellent device performance. There is reason to be cautious here, as it has been found in recent years that multiple morphologies, some hierarchical, and some not, can produce high PCE in the same materials system. One cannot prove that the hierarchical morphology is the origin of the performance without producing a morphology that is the same in every other way (same crystallinity, same molecular orientation, same aggregation state, etc) but not hierarchical. This would be really tough to do, perhaps impossible, and I would not ask the authors to attempt it.

We can accept that the morphology was optimized, that the optimized morphology is hierarchical, and that the optimization led to high device performance. But in the absence of more evidence, the authors must change the strong assertions about the role of the morphology to speculative versions, and the paper would then be suitable for publication. Some of the assertions that are too strong are listed below:

- "All these phenomena consequently contributed to effective photocurrent" [line 230] - no proof. Only some of the phenomena might be required.

- "due to the optimized hierarchical morphologies, balanced electron and hole mobility were observed for both systems." [line 241] - can only be proven by showing that balanced hole and electron mobilities are not balanced if the hierarchical trait is removed from the morphology.

- "In this study, however, the existence of hierarchical morphologies balanced the charge separation and charge transport." [p282] - similar concerns

Response to the comments of reviewer

We would like to thank the referees for spending time on this paper and providing invaluable comments which substantially helped to improve the quality of the paper. The manuscript has been carefully revised according to the comments.

Reviewer #3 (Remarks to the Author):

General comments: The authors have re-characterized the morphology discussion entirely, embracing my suggestion that the observed morphology should be classified as hierarchical, and also adopting most of my other suggestions. I am satisfied that my previous round of concerns have been mostly addressed. Half of the paper is rewritten, and there is a lot of new prose to evaluate.

My principal concern about the new prose is that it is too strong in its assertions about the role of the hierarchical morphology in producing the excellent device performance. There is reason to be cautious here, as it has been found in recent years that multiple morphologies, some hierarchical, and some not, can produce high PCE in the same materials system. One cannot prove that the hierarchical morphology is the origin of the performance without producing a morphology that is the same in every other way (same crystallinity, same molecular orientation, same aggregation state, etc) but not hierarchical. This would be really tough to do, perhaps impossible, and I would not ask the authors to attempt it.

We can accept that the morphology was optimized, that the optimized morphology is hierarchical, and that the optimization led to high device performance. But in the absence of more evidence, the authors must change the strong assertions about the role of the morphology to speculative versions, and the paper would then be suitable for publication. Some of the assertions that are too strong are listed below:

- "All these phenomena consequently contributed to effective photocurrent" [line 230] - no proof. Only some of the phenomena might be required.
- "due to the optimized hierarchical morphologies, balanced electron and hole mobility were

observed for both systems." [line 241] - can only be proven by showing that balanced hole and electron mobilities are not balanced if the hierarchical trait is removed from the morphology.

- "In this study, however, the existence of hierarchical morphologies balanced the charge separation and charge transport." [p282] - similar concerns

Author response: We would like to thank the referee for their helpful comments. We do agree with your comments about the role of morphology, as systems with both, the hierarchical morphology and the nanofibril morphology can produce high PCEs. However, due to the structures and compatibility of the involved molecules, some systems prefer to form the hierarchical morphology over the nanofibril morphology, and vice versa. Likewise, and as you have rightfully mentioned, fabricating two systems that are same in every way (same crystallinity, same molecular orientation, same aggregation state, etc.) except the nano-morphology, is a fairly difficult task as we have already tried various conditions (D/A ratios, thermal annealing and solvent annealing, etc.) while device optimization.

Therefore, the article has been seriously revised to describe the function of the active layer morphology with speculative versions.

We have changed the description from "All these phenomena consequently contributed to effective photocurrent, indicating that the nano-structural characteristics with multiple length scales as well as coexisting crystals are key factors for high performance" to "The presence of these optimized hierarchical morphologies indicated that the nano-structural characteristics with multiple length scales as well as coexisting crystals are among key factors for high performance."

We have changed the description from "due to the optimized hierarchical morphologies, balanced electron and hole mobility were observed for both systems." to "these results reflect that the hierarchical morphologies of both systems are efficient for charge transport."

We have changed the description from "In this study, however, the existence of hierarchical morphologies balanced the charge separation and charge transport, and therefore high PCE was obtained." to "In this study, however, the existence of hierarchical morphologies were important for

the charge separation and transport, and ultimately led to a high PCE.”